# Occurrence and distribution of anthropogenic persistent organic pollutants in coastal sediments and mud shrimps from the wetland of central Taiwan

**Shagnika Das[1,2], Andres Aria[3,4], Jing-O Cheng[5], Sami Souissi[2], Jiang-Shiou Hwang[1,6], Fung-Chi Ko [5,7] ***

**1** Institute of Marine Biology, National Taiwan Ocean University, Keelung, Taiwan, **2** University Lille, CNRS, University Littoral Cote d'Opale, UMR 8187, LOG, Laboratoire d'Océanologie et de Géosciences, Wimereux, France, **3** Argentine Institute of Oceanography, Bahia Blanca, Argentina, **4** National South University, Chemistry Department, Area III, Bahía Blanca, Argentina, **5** National Museum of Marine Biology and Aquarium, Checheng, Pingtung, Taiwan, **6** Center of Excellence for the Oceans, National Taiwan Ocean University, Keelung, Taiwan, **7** Institute of Marine Biology, National Dong-Hwa University, Pingtung, Taiwan

* ko@gms.ndhu.edu.tw

**Data Availability Statement:** All relevant data are within the manuscript and its Supporting Information files.

## Abstract

Sediment profile and mud shrimp (*Austinogebia edulis*) from the coastal wetland of central Taiwan in 2017 and 2018 were analyzed for concentration, source, and composition of persistent organic pollutants (POPs) including polycyclic aromatic hydrocarbon (PAHs), polybrominated diphenyl ethers (PBDEs), organochlorine pesticides (OCPs; DDT and HCB), and polychlorinated biphenyls (PCBs). Sediment profiling indicated PAH concentrations reaching 254.38 ng/g dw in areas near industrial areas and PAH concentrations of 41.8 and 58.42 ng/g dw in sampling areas further from industrial areas, suggesting that the determining factor for spatial distribution of POPs might be proximity to contaminant sources in industrial zones. Based on molecular indices, PAHs were substantially of both pyrolytic and petrogenic origins. The main sources for PCBs were Aroclor 1016 and 1260 and the congener BDE-209 was the dominant component among PBDE congeners. While we were unable to obtain live mud shrimp samples from the heavily contaminated areas, in samples from less contaminated areas, the risk assessment on mud shrimp still illustrated a borderline threat, with DDT concentrations almost reaching standardized values of Effects Range-Low (ERL). Bioaccumulation factors for DDTs and PCBs (17.33 and 54.59, respectively) were higher than other POPs in this study. Further study is essential to assess and understand the impact of these chemicals on the wetland ecosystem near this heavily industrialized area.

## 1. Introduction

Persistent organic pollutants (POPs) are pervasive contaminants found in different domains of the environment [1–4], especially the coastal and esturaine sediments due to a rise in

**Funding:** We acknowledge a grant of the Ministry of Science and Technology (MOST) of Taiwan for financial support through grants: MOST 104-2611-M-019- 004, MOST 105-2621-M-019 001, MOST 105-2918-I-019-001, MOST 106-2621-M-019-001, and MOST 107-2621-M-019 001. The funders had no role in study design, data collection and analysis, decision to publish, or preparation of the manuscript.

**Competing interests:** The authors have declared that no competing interests exist.

industrialization along the coastal regions. Increasing anthropogenic activities, such as discharge of pesticides and fertilizers and incomplete combustion of petrochemicals, have significantly elevated the loading of POPs to the underlying sediment, which has often been considered as the ultimate sink for numerous chemical pollutants [5–11]. Coastal wetland with intertidal zones are considered to receive the most stress due to the fluctuation of different physical parameters, and are more susceptible to anthropogenic activities [12, 13]. Thus, contamination of tidal flats in wetland is a common scenario and measures should be taken to protect the wetland ecosystem.

The vast intertidal flat of coastal zone at Changhua County, Taiwan, represents a complex environmental ecosystem with high biodiversity of crabs, shrimps, fish, and clams. Mud shrimp (*Austinogebia edulis*) is the dominant species in the tidal flat, where they build deep burrows (>1 m) as their habitats. This mud shrimp is prominent in the Changhua County and is considered to be ecologically and commercially essential; hence, it is frequently monitored through different governmental programs. A previous study demonstrated the sensitivity of *A. edulis* to trace metals and alterations to enzymatic activities and physiology caused by higher concentrations of cadmium [14]. During the past decade, the *A. edulis* population has extensively declined for unknown reasons. Since benthic organisms are more likely to get exposed to contaminants in sediment than in air and water, the sediment is a good index for recording contamination levels. An emerging hypothesis is that *A. edulis* experience higher contaminant stress since their burrows are commonly high in organic matter compared to surrounding sediment and can therefore accumulate more POPs [15].

Although many POPs such as polychlorinated biphenyls (PCBs), polybrominated diphenyl ethers (PBDEs), and organochlorine pesticides (OCPs) have been prohibited in many countries, they still continue to be reported as toxic chemicals to pose a global threat to the ecosystem. In 1988, after several years of extensive usage and application, the Environmental Protection Administration of Taiwan listed 10 POPs as toxic substances [16]. Despite this regulation, previous research has demonstrated extensive levels of POPs in Taiwan, including PAHs, PBDEs, and OCPs, mostly in rivers and coastal areas such as Gao-Ping River [17–19], Kenting coral reef [20, 21], Danshui River and Keelung River [22–25]. Given this precedence, there are few studies on POP concentrations in the western coast of central Taiwan, an area heavily impacted by anthropogenic activities [26]. In particular, the Changhua coastal industrial park, also known as the Changbin Industrial Park, is a conglomeration of industries, which includes a variety of chemical industries, processors for metals, textile, and food production. Previous research in the area has demonstrated poor air quality with elevated levels of trace metals and PAHs [27, 28]. However, there is no information on the distribution and effects of POPs in the Changbin Industrial Park.

The aims of this study are to investigate the spatial and temporal distribution of several POPs in sediments and benthic organisms (mud shrimp, *A. edulis*) near Changbin Industrial Park, identify the possible sources of POPs, and assess the potential ecological risks to benthic organisms in this area.

## 2. Material and methods

### 2.1 Sample collection

The sediment and biota (mud shrimp) samples from around the industrial area in the western coast of central Taiwan were collected at five stations along the coastline: stations A, B, C, D, and E (Fig 1; S1 Table) from September 2017 to January 2018. The sampling sites experience a wet season (monsoon) from late July until the end of October and a dry season from December until March (winter). Sampling was conducted once in the wet season (September, 2017;

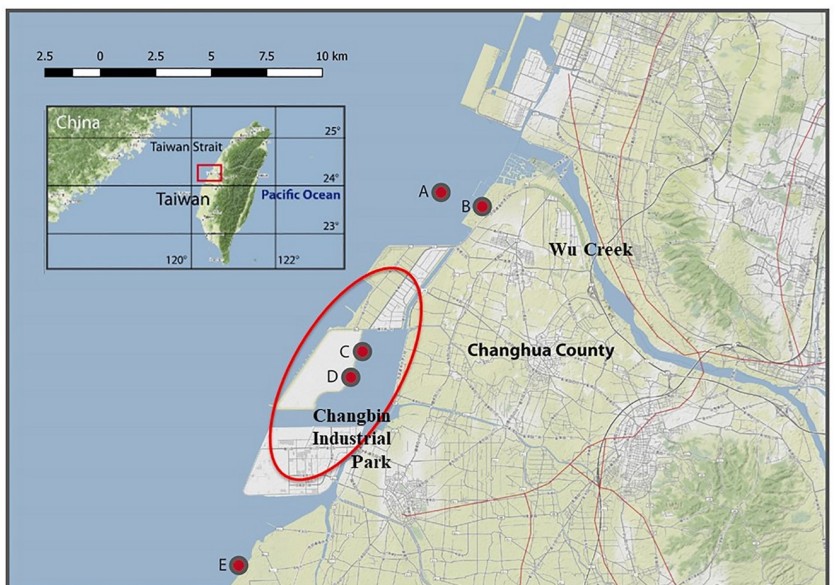

**Fig 1. Map showing the sampling stations (A, B, C, D and E) at the western coast of Taiwan.**

temperature of ~28˚C) and once in dry season (January 2018; temperature of ~15˚C). The sediment samples were collected by coring with the deepest layer at 50 cm, middle layer at 25 cm, and the surface at 0 cm. Some sediment samples were kept for analysis of total organic carbon (TOC) and grain size. In this study, the mud shrimps *A. edulis* were collected only from station A and E due to their absence from the other stations (B, C, and D) during sampling. Shrimp samples were washed by ambient water to remove the sand on exoskeleton, wrapped in aluminum foil, stored in an icebox, and immediately transported to the laboratory. Sediment samples were homogenized on a clean stainless-steel tray, transferred to solvent cleaned glass bottles, and all samples were stored in -20˚C until further analysis. *A. edulis* is not defined as threatened or protected in Red List for Species. As the sampling was conducted outside of national parks or any protected area, no specific permissions were required.

## 2.2 Sample preparation

The preparation and extraction of sediment samples and mud shrimp (*A. edulis*) were conducted as described by previous studies [20, 21]. Briefly, about 5 g of dried sediment and shrimp of each sample were ground finely with anhydrous $Na_2SO_4$ in a mortar pestle, following by extraction in a Soxhlet apparatus for 24 h with hexane and acetone (1:1; v/v). Every sample was spiked with surrogate standards (*d8*-Napthalene, *d10*-Fluorene, *d10*-Fluoranthene and *d12*-perlene for PAHs and PCB congener14, congener65 and congener166 for PCBs, OCPs, and PBDEs) prior to extraction in order to calculate the percentage of procedural recovery. Activated copper wires or granules were used as necessary to remove elemental sulfur from sediment samples. A rotary vacuum evaporator was used to concentrate the extract. Fractionation and further cleaning of the extract to remove polar interference was performed using 8 g of 6% deactivated alumina packed in a column. Extracts were concentrated using a rotary evaporator followed by a gentle purified nitrogen stream. PAH concentrations were measured by gas chromatography-mass spectrometer (GC-MS) (Varian 320). After PAH analysis, the sample extracts were passed through a glass column packed with 8 g of 2.5% deactivated florisil and covered with about 1 cm $Na_2SO_4$. The column was conditioned with 35 ml petroleum

ether (PE) and dichloromethane (DCM) mixture (1:1) solvent with further addition of 35 ml PE. The PCBs, OCPs, and PBDEs were eluted with 35 ml PE. The eluate was transferred to hexane, concentrated by rotary evaporation, and further dried to less than 0.1 ml under a gentle stream of nitrogen. For quantification, the internal standards (PCB congener30 and congener204 for PCBs and OCPs, and PBDE congener209 for PBDEs) were added to each sample before analyzing the PCB, OCP, and PBDE concentrations with a GC-triple- quadruple mass spectrometry system (TQ-8050, Shimadzu).

## 2.3 Analytical techniques

The PAHs were determined using a Varian 320 GC-MS, while PCBs, DDT, HCB and PBDEs analysis were conducted via GC-MS/MS (TQ-8050, Shimadzu). Separation was performed with a VF-5ms column (30 m, 0.25 mm i.d., 0.25 μm film thicknesses) for PAHs, and OCPs, VF5ht (15 m, 0.25 mm i.d., 0.1 μm film thickness) for all PBDE congeners, and DB-5 (60 m, 0.25 mm i.d., 0.25 μm film thickness) for PCBs. The oven temperature programs for POPs of the instrumental analysis are described in supplementary data (S1 Protocol). The method detection limits (MDLs), recovery analyzed process, and the quantitative ion and confirm ion in mass spectrometry for each POP are listed in S2–S4 Tables, respectively.

## 2.4 Total organic carbon determinations

The total organic carbon (TOC) content in the sediment was determined by using an Elementar Vario EL III (Exeter Analyutical Inc, Germany) after removal of inorganic carbon. [19, 21]. For each sample, three replicates were used to determine the TOC.

## 2.5 Quality assurance and quality control

Analysis was conducted according to the standard quality assurance protocols. Glassware and apparatus were washed with detergent and double distilled water, followed by a rinse with acetone and hexane. All analysis included procedural blanks, samples (in duplicate), and spiked samples to assure the quality of the extraction and to detect any transmission of contaminants that might have occurred during analysis. In order to find out the recovery efficiency, four PAH surrogates (*d8*-napthalene, *d10*-fluorene, *d10*-fluoranthene, and *d12*-perylene) and three PCB surrogates (congener14, congener65, and congener166) were added prior to extraction in all the sample tubes, including the blank. The average recoveries of the standards were 62.8 ±13.5%, 84.2±10.7%, and 89.9±12.4%, 82.9±18.3% for the PAHs and 73.9±14.5%, 83.3±16.8%, 84.0±17.3% for the PCBs, respectively. The concentrations of POPs were not corrected for surrogate recoveries. The ranges of recovery of spiked individual POPs were 65.3% -100.6% for PAHs, 57.2%-90.6% for PCBs, 64.4%-80.3% for HCB, 64.4%-94.6% for DDTs, and 66.6%-118.8% for PBDEs, respectively. The method detection limits (MDLs) of POPs were defined as the average mass of each compound in the blanks plus three times the standard deviation. The mass of compounds below the MDLs were computed as zero. Quantification of POPs was done by the internal standard method. The relative standard deviation of relative response factor (RRF) was below 10%.

## 2.6 Calculation of bioaccumulation factors (BAFs)

In order to implement a tool for the regulation of contaminated habitats, and for the assessment of potential risk to benthic animals, calculation of bioaccumulation factors (BAFs) have been proved to be a useful approach. The BAFs for each pollutant in stations A and E was

calculated by using the Eq (1).

$$BAFs = \frac{Concentration\ of\ POPs\ in\ mud\ shrimp}{Concentration\ of\ POPs\ in\ sediment} \tag{1}$$

## 3. Results and discussion

### 3.1 Spatial and temporal distribution of POPs

Among the five POPs series that were analyzed, PAHs, PCBs, DDTs and PBDEs were detected in all the samples except HCB at station A and station E (Fig 2a). Station C had the highest concentration of t-PAHs (332.95 ng/g dw; dry weight), t-PCBs (4.34 ng/g dw), t-HCBs (0.18 ng/g dw) and t-PBDEs (58.36 ng/g dw) and station D had the highest concentration of t-DDTs (2.63 ng/g dw). Station C and station D had much higher concentrations of all the measured POPs, indicating site-specific pollution levels at these stations, probably due to their proximity to Changhua Industrial Park. The POP concentrations in this study did not significantly differ between seasons. Fig 2b shows the total concentration of all the POPs in the sediment core depths (0 cm, 25 cm, and 50 cm) at the five sampling stations were not significantly different. Fig 2a shows the concentration of different POPs in the surface sediment of five sampling stations around Changhua County collected in the wet season (August) and the dry season (January).

### 3.2 Comparison of POP concentrations with other coastal and estuarine regions in the world

The concentration of PAHs, PCBs, and DDT in the present study was in a range similar to that of Gao-ping estuary in south Taiwan and in Southeast Asian countries, which is substantially

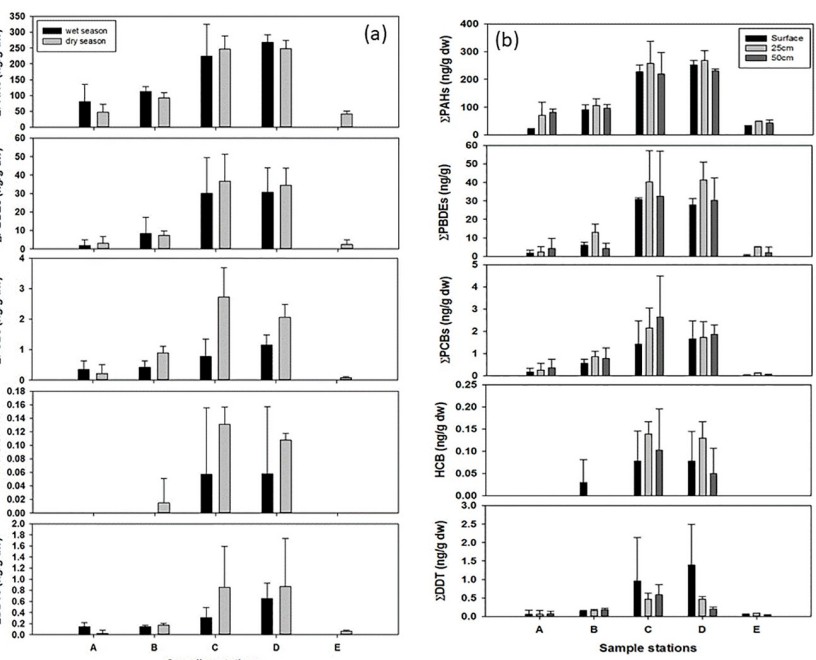

**Fig 2.** Total concentrations of POPs in different stations (A, B, C, D, E); a. Seasonal variation. b. distribution at different depths (0 cm, 25 cm, and 50 cm).

**Table 1. Comparison of the POP concentrations (ng/g dw) in the sediments from the other coastal areas with the present study.**

| Locations | t-PAH (ng/g dw) | t-PCB (ng/g dw) | t-DDT (ng/g dw) | References |
|---|---|---|---|---|
| **Stations A, B, and E** | **19.96–129.66** | **0.04–1.11** | **0.03–0.21** | **This study** |
| **Stations C and D** (Changbin Industrial Park) | **135.15–332.95** | **0.24–4.34** | **0.11–2.63** | **This study** |
| Pearl River Estuary, China | 138–1100 | - | 1.38–25.4 | [29] |
| Yangtze Estuarine, China | 32.10–71.10 | ND-63 | ND-5.10 | [30] |
| Victoria Harbor, Hong Kong | | 3.2–27 | 1.4–30 | [31] |
| Coastal region, Singapore | 12.65–93.85 | 1.40–330 | 3.40–46.10 | [32] |
| Coast of Korea | 9.1–1400 | 0.17–371 | 0.01–135 | [33] |
| Gao-ping Estuaries, Taiwan | 1.43–356 | 0.38–5.90 | 0.44–1.88 | [34] |
| Northeastern coast of India | - | 0.18–2.33 | 0.18–1.93 | [35] |
| Bay of Bengal, India | 20.35–2615.38 | 0.02–6.57 | 0.04–4.79 | [36] |
| West coast of India | - | - | 1.47–25.17 | [37] |
| Bay of Biscay, France | 0.7–300 | ND-375 | - | [38] |
| The Baltic Sea | 9.5–1900 | 0.01–6.20 | 0.13–0.50 | [39] |
| Cantabrian Sea, Spain | 19–2123 | ND-160 | 4.2–25 | [40] |
| San Francisco Bay, CA | 2944–29590 | - | 11–23330 | [41] |
| Guánica Bay, Puerto Rico, USA | 0.64–4663 | 140.11–3059.90 | 0.00–69.25 | [42] |
| Bahia Blanca Estuary, Argentina | 15–10260 | 0.61–17.6 | ND-2.3 | [43] |

ND: lower than MDLs

lower than in the other coastal areas of the world (Table 1). In this study, the DDT concentrations near the Industrial Park (stations C and D) were significantly higher than in other parts of Changhua County, but much lower when compared to the DDT concentrations found across the globe. In this study, we also analyzed the concentration of PBDEs. The t-PBDE concentrations ranged from 0–58.36 ng/g dw. This range of t-PBDEs was similar to the Macao coast in China where the reported range was from 0.6–41.3 ng/g dw [44]. We found that the discharge of PBDEs in the Industrial Park was also much higher than in other parts of Changhua County, including the southwestern coast (station E) where the t-PBDE concentrations was below the detection limits.

### 3.3. Compositional profiles

**3.3.1. PAHs.** PAH isomer ratios have been used to determine PAHs sources and estimate the importance of combustion and petroleum-derived PAHs [45, 46]. The index of combustion (anthropogenic) input of PAHs is an increase in the proportion or molecular mass totals of the less stable or kinetically produced parent PAH isomers relative to the thermodynamically stable isomers, such as fluoranthene relative to pyrene. Index calculations are traditionally restricted to PAHs within a given molecular mass to minimize factors such as differences in volatility, water/carbon partition coefficients, adsorption, and in most cases closely reflect the source characteristics of PAHs.

PAH rations considered were primarily proportions of fluoranthene to fluoranthene plus pyrene (Fl/202) and indeno[*1, 2, 3-cd*]pyrene (IP) to IP plus benzo[*ghi*]perylene (IP/276) (Fig 3). Fl/202 ratios less than 0.40 indicate petroleum (oil, diesel, and coal) and ratios between 0.40 and 0.50 indicate liquid fossil fuel (vehicle and crude oil) combustion, while ratios over 0.50 indicate grass, wood or coal combustion. Similarly, IP/276 ratios less than 0.20 indicate petroleum, ratios between 0.20 and 0.50 indicate liquid fossil fuel combustion, and ratios over 0.50

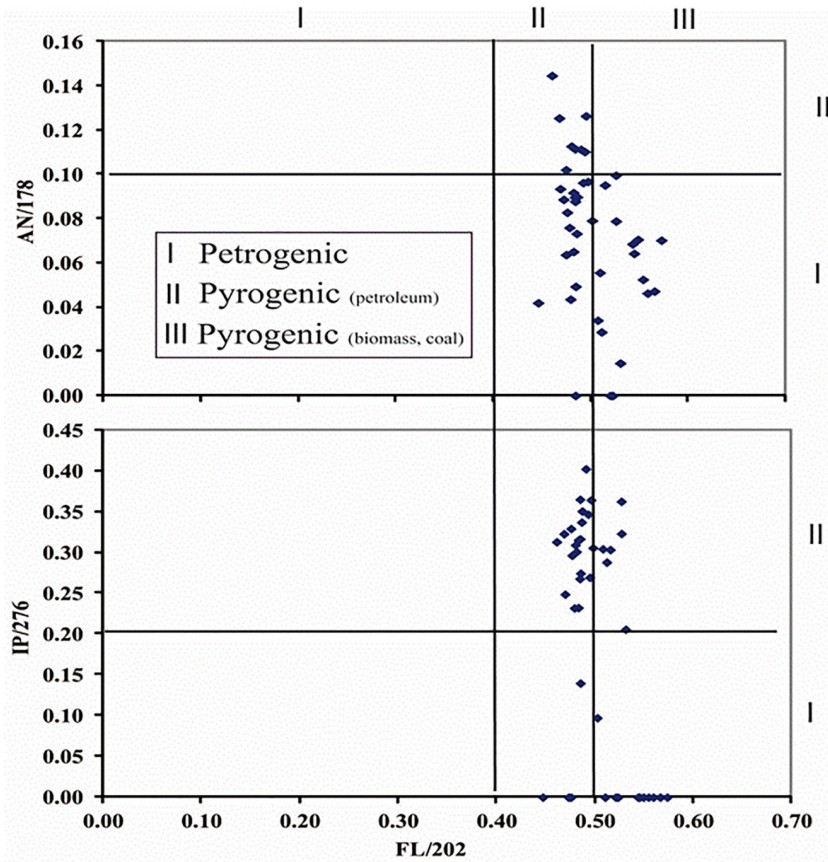

**Fig 3. Comparison of selected PAHs ratios for 42 sediment samples along the area of study.** Abbreviations refer to the ratios of fluoranthene plus pyrene (Fl/202), indeno[*1, 2, 3-cd*] pyrene (IP) to IP plus benzo[*ghi*]perylene (IP/276) and anthracene (An) to An plus phenanthrene (An/178).

indicate grass, wood or coal combustion. Furthermore, these two parent PAH ratios are supplemented by anthracene (An) to An plus phenanthrene (An/178). An/178 ratios less than 0.10 indicate petroleum, while ratios greater than 0.10 indicate combustion.

In this study, sediment samples presented a mean Fl/202 ratio of 0.50 ± 0.03 (n = 42) and an IP/276 mean ratio of 0.25 ± 0.12 (n = 35) (Fig 3), indicating a pyrogenic source impacting the area. The mean An/178 ratio was 0.07 ± 0.03 (n = 42), pointing to petrogenic inputs.

Principal Components Analysis (PCA) was used to extract underlying common factors (principal components, PCs) and to analyze relationships among observed variables. As a result of an effective extraction process, PC1 accounted major proportion of the total data variance while the second and following PCs progressively explained smaller amounts of data variation. Prior to analysis, values under the method detection limits (MDLs) in the data set were replaced with random values under the MDL value.

Concentrations of 40 PAHs as active variables and 42 samples as cases were used. The number of factors extracted from the variables was determined according to Kaiser´s rule, which retains only factors with eigenvalues exceeding 1. As performed in other studies [47] a method of factor rotation to get as many positive loadings as possible to achieve a more meaningful and interpretable solution was preferred (Varimax normalized).

The majority of the variance (88.3%) was explained by three principal components vectors. PC1 accounted for 73.1% of the total variance, PC2 accounted for 6.2%, while PC3 accounted

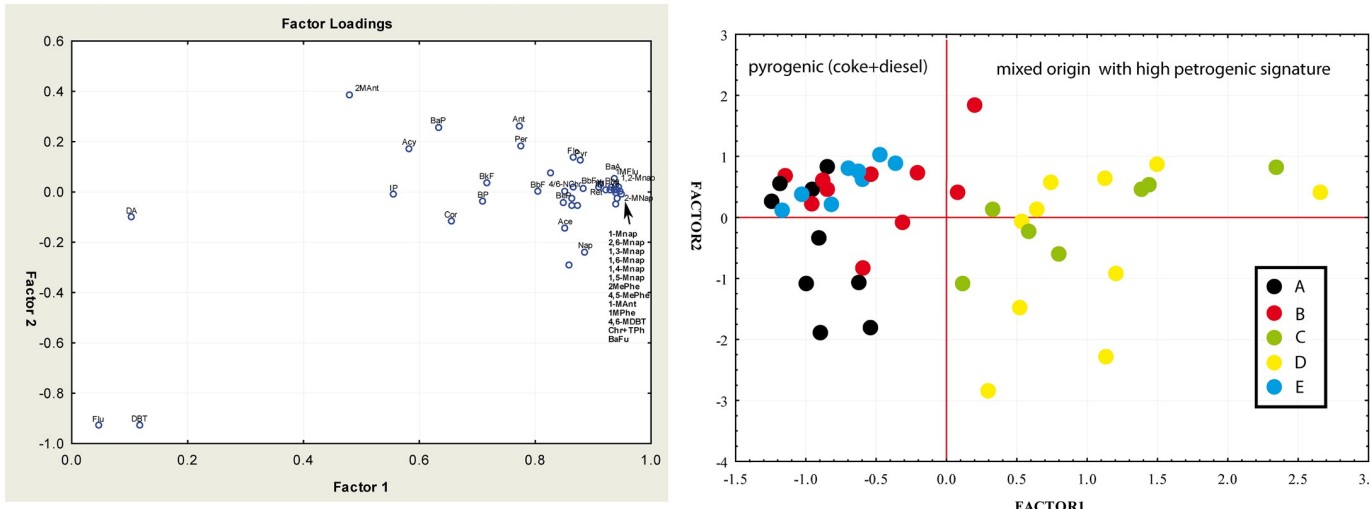

**Fig 4.** a. The PCA loading plot of sedimentary PAHs; b. Score plot illustrating the distribution of PAHs compounds in the sampled areas along PC1 and PC2 axis.

for 3.9% of the variance. Fig 4a shows the loadings for the individual PAHs at the principal components plot. All the compounds were found to be positively correlated along the PC1 axis; indeed, PC1 had high correlation ($r^2 = 0.7$) with alkylated derivatives of PAHs, which are markers of petrogenic origin. This component also included some pyrogenic markers such as fluoranthene, pyrene, benz[*a*]anthracene, chrysene (coal combustion) and even retene, a marker of wood combustion. In addition, PC1 compounds were strongly correlated with the t-PAHs concentration ($r^2 = 0.91$), indicating PC1 as mixed origin with over-imposition of petrogenic origin and a quantitative correlation component.

PC2 presented significant positive loadings for two 3-ringed PAHs compounds: fluorene and dibenzothiophene. Fluorene has been reported as a dominant PAH in the coke oven signature [48] while dibenzothiophene (thiophenes in general) is a marker of diesel-powered vehicles [49]. Such PAHs are the result of combustion/pyrolitic processes and are absent in crude oil or refined products. Consequently, PC2 was defined as a pyrogenic component including coke combustion and diesel motors exhaust.

PC3 presented a significant correlation with indeno[*1, 2, 3-c,d*]pyrene, a six fused ring compound, which is a common marker of pyrolysis. The principal components plot (Fig 4a) shows different PAHs clustering. PAHs in the three main clusters may be originated from different origin sources. Following PCA, a portion of the total variance of PAH concentrations is explained by source contribution, with petrogenic origin as the prevalent contribution over the sampled area. These results are consistent with our previous findings. Fig 4b shows the 2-D score plot with PC1 and PC2 axes, characterizing the sampling stations according to the first and the second components. In this context, we identified three major groups of samples distributed along the three axes (Fig 4a). PC1-positive coordinates include 19 samples dominated by stations C and D. These samples are located in a petrochemical industrial park wastewater discharge zone, an area defined above as a hotspot because of its total PAHs levels. The remaining samples can be divided into different PC2 and PC3 contributions, with stations B and E mainly correlating with PC2 (coke + diesel combustion) and stations A and B correlating with PC3 (gasoline, coal combustion). In brief, PCA allowed us to assess different PAH sources and classify the sampling sites. These PCA results reflect the conclusions drawn from the ratio metric analysis.

As shown in Fig 4b, stations C and D represented marked petrogenic PAH inputs, concluding that the PAHs sources at the most impacted locations in Changhua County are mainly petrogenic inputs and not combusted oil and petroleum derivatives.

**3.3.2. Organochlorines and PCBs.** Commercial grade DDT generally contains 90% DDT, 5% DDE, <1% DDD, and <0.5% unidentified compounds [50]. DDT-isomers have a long persistence in the environment, gradually degrading to DDE and DDD under both aerobic and anaerobic conditions. In general, the trend of concentrations of DDT and its metabolites present in sediments was DDT>DDD/DDE, indicating quite recent inputs of commercial DDT to the environment. Despite this, different spatial trends were identified: in the north (stations A and B; catchment of city sewage outlet and harbours), there were medium DDT levels (0.17 ng/g dw) and the trend observed for its derivatives was DDT >DDD> DDE. In the south (stations D and E; reserve), the DDT levels were the minimum recorded (0.02–0.06 ng/g dw), while the industrial stations C and D showed the maximum levels (>0.8 ng/g dw). The dominance of DDTs in stations C and D sediments, as well as the maximum concentration achieved in top layer, indicates slow degradation of DDTs or recent inputs of DDT at these locations [51].

Concentrations of PCBs in worldwide comparison were relatively lower (< 4 ng/g dw). PCB levels were highest at stations C and D, lower at stations A and B, and lowest at station E, reflecting the land use/cover in these areas. The major compounds found in the area were congeners 132,153, 31, 28 and 5+8. Comparing the pattern of percentage of chlorinated compounds with the average known Aroclor mixes (UNEP), we found that the source of PCB contamination in the study area was mixed pattern, involving Aroclor 1016 and Aroclor 1260.

**3.3.3. PBDEs.** The concentration of total PBDE (Σ10 congeners) ranged from 0–57.6 ng/g dw, with BDE-209 being the most abundant congener at all the stations and in both seasons. The highest concentration of PBDE was found at station C (57.60 ng/g dw) in the deepest layer (50 cm) of sediment in the dry season (January). This result was in accordance with studies in other parts of world which described greater concentrations of POPs in dry seasons than in the wet seasons and higher PBDE concentrations near chemical industries [25, 52, 53]. BDE-209, the most abundant congener, composed almost 85–90% of the total PBDE concentration in all the stations in both seasons. The highest BDE-209 concentrations were recorded in station C (57.60 ng/g dw at 50 cm in dry season and 47.89 ng/g dw at 25 cm in wet season). Similar results have been published by previous studies, where BDE-209 was found to be the most abundant congener, accounting for almost 90–100% of the composition of t-PBDE and showing highly variable concentrations throughout the world [25, 53, 54]. The observation is corroborated by recent studies demonstrating the high commercial usage of Br-10 mixtures in Taiwan. The low brominated BDEs found in the top layer of each station might be the product of debromination of higher brominated congeners of PBDE.

## 3.4. Concentrations of POPs related with sediment characteristics

The total organic carbon (TOCs) of sediment is often considered as a prime factor for the distribution of POPs in a particular area and are widely compared in studies related to organic contaminants [22–25, 55, 56]. Persistence of POPs in aquatic sediments is due to their low rate of degradation and vaporization, low water solubility, and high partitioning to particles and organic carbon. To test this in the present study, a correlation between each POP level and % TOC was assessed in this study (Fig 5).

The TOCs of the sediments ranged from 0.16–0.93%. The lowest TOC content was recorded at station A and the highest at station C, both in the middle layer (25 cm) of the sediment column. There were strong and significant correlations between each POP (t-PAH, t-

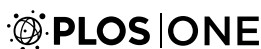

**Fig 5. The relationship between TOC (%) and t-PAH, t-PCB, t-DDT, and t-PBDE in ng/g dw.**

PCB, t-DDT and t-PBDE) with TOC (Fig 5). Previous studies have reported similar findings when comparing TOC and POP concentrations [25, 34, 57, 58, 59]. Hence, TOC content can be considered as a useful tool to assess or surveil the concentrations of diverse organic contaminants in sediments [25, 57,58].

## 3.5. Ecological risk assessments of POPs in sediments

Taiwan still lacks a standard for acceptable concentrations of individual POPs in seafood and sediment. However, previous studies have used the general rule presented by [60] from the Canadian councils of Ministers of the Environment [61], which uses the Threshold Effect Level (TEL) and the Probable Effect Level (PEL) to estimate the risk of POPs in sediments and benthic organisms. This standard criterion represents the limit above which (50% frequency, "Effects Range-Median") contaminant concentration will be considered toxic and below which (10% frequency, "Effects Range-Low"), contaminant concentrations will rarely cause detrimental effects. Using these sediment quality guidelines (SQG) set by [60], none of the POP concentrations at the stations exceed the ERL values (Table 2). However the t-DDT concentrations at stations C and D are closer to the standardized ERL values. This can be explained by the use of pesticides for nearby agriculture and the proximity of these two stations to the Industrial Park. Also, the concentration of t-PAHs and t-PCBs are higher at stations

**Table 2. Comparison of standard values set for toxicity (SQG and CCME values ng/g dw) with the POP concentrations in the sediments from this study.**

| | SQG [60,61] | | | | This study | | | | |
|---|---|---|---|---|---|---|---|---|---|
| | | | | | Stations | | | | |
| | ERM | ERL | PEL | TEL | A | B | C | D | E |
| t-PAHs | 44792 | 4022 | 6676 | 655 | 58.42 | 99.2 | 238.93 | 254.38 | 41.80 |
| t-PCBs | 180 | 22.7 | 189 | 22 | 0.25 | 0.73 | 2.07 | 1.75 | 0.07 |
| t-DDTs | 46.1 | 1.58 | 4.77 | 3.89 | 0.06 | 0.16 | 0.67 | 0.79 | 0.06 |

C and D than at the other stations, which may have caused the absence of mud shrimps at these sites. Therefore, the concentrations of POPs in the wetland sediment of this study, including the Changhua Industrial Park, likely have rudimentary or marginal effects on benthic organisms.

### 3.6 Bioaccumulation of POPs in the mud shrimp

*Austinogebia edulis* is a common seafood consumed by locals of the western coast of Taiwan and it has high economic importance. The concentration of each measured POP in *A. edulis* has been listed in Table 3. Out of the five stations (A, B, C, D, and E) only two stations (A and E) showed the presence of mud shrimps. The concentration of t-PAHs was higher when compared to the other POP concentrations. HCB was not detected in any of the samples that were tested. The concentration of all the POPs tested were found to be much higher at station E than station A. Since the Taiwanese government has declared station A a restricted mud shrimp conservation area, the level of contamination by organic pollutants are comparatively lower than in surrounding unrestricted and open areas.

The BAFs in the mud shrimp *A. edulis* were much higher at station E than at station A, especially for t-PCBs and t-DDTs (Table 3). *Austinogebia edulis* are burrowers in the muddy substratum and are get exposed to the sediment column throughout their entire lifespan. Previous literatures have documented that BAFs values are highly variable and dependent on various parameters, like direct contact or exposure of the organisms with the sediment column, amount of the residue present in the sediment etc. [62–66]. Thus, even a hydrophobic compound like PCB may have a higher BAF value in the body of a burrowing mud shrimp. Previous literature noted ranges of BAFs for PAHs to be higher in certain crabs and lugworms than in oysters and clams [64]. Several studies have accounted for varying BAFs for PCB and PBDE in fishes, crabs, mussels, shrimps, etc. For instance, the Chinese Mitten Crab exhibited the highest value of Biota-Sediment Accumulation Factor (BSAF) for PCBs (2900), followed by the shore crab in the Scheldt estuary of Netherlands–Belgium (2330) [67]. At the same site the

**Table 3. The total concentrations and bioaccumulation factors (BAFs) of POPs in mud shrimps.**

| Station | t-PAHs | t-PCBs | t-PBDEs | t-DDTs | HCB |
|---|---|---|---|---|---|
| | | | Concentration (ng/g) | | |
| Station A | 61.9 | 0.5 | 1.2 | 0.4 | nd |
| Station E | 94.1 | 3.9 | 2.2 | 1.1 | nd |
| | | | Bioaccumulation factor | | |
| Station A | 1.1 | 1.9 | 0.5 | 6.2 | |
| Station E | 2.3 | 54.6 | 0.9 | 17.3 | |

nd: lower than MDLs

value of PCB BSAF in the brown shrimp (606), blue mussels (1090), and worms (1180) were found to be comparatively lower [67]. However, the PCB BSAF for the Blue crab (0.22–1.7) and the white perch (0.34–1.5) in the Passaic River, USA accounted for minute values of bioaccumulation factor [65]. The BSAF for PBDEs in the Scheldt estuary was also higher in Chinese mitten crab (2520) and shore crab (598), but in comparison was much lower in brown shrimp (162) and blue mussels (304). The PBDE BSAF found in the northern horse mussel of the Vancouver Island, Canada varied from 0.95–527 [68]. The present study found lower BAF values for each measured POPs in mud shrimps when compared to other global studies. The variations in the results can be accounted for by different trophic levels or the modes of ingestion amongst the benthic biota. For instance, lower molecular weight PAHs such as phenanthrene and anthracene have been reported to absorb compounds from the interstitial water directly, whereas higher molecular weight compounds adsorb on particulate matter [64, 69]. Thus, the bioaccumulation pathways for different aquatic and intertidal organisms differ greatly depending on the molecular weight of the compounds, modes of ingestion, absorption through skin, and the ability of the organisms to metabolize the compounds.

## 4. Conclusion

This study presents the first comprehensive survey of PAHs, PBDEs, OCPs, and PCBs in sediments from Changhua County, Taiwan providing useful information on concentrations, composition and sources. The spatial distribution of POPs showed that proximity to sources was the most important determining factor for the distribution of these contaminants. In general, POP concentrations were greater in samples collected near the industrial area (C and D) than those from the non-industrial locations (A, B, and E). Molecular indices such as Fl/202, IP/276, and An/178 revealed the existence of both pyrolitic and petrogenic inputs at the area. Further, the use of PCA enabled the classification of sampling sites in accordance to their main source of PAHs. Considering PCBs, Aroclor 1016 and 1260 were assessed as the main technical sources for the area. Although levels were below scientific sediment guidelines, the study identified recent DDT inputs to the area. Beyond the anthropogenic impact on the sediment, POPs appear to pose a potential rudimentary or marginal risk in regards to their effect on benthic organisms.

## Supporting information

**S1 Table. Sampling stations around the Changhua Industrial Park along the western coast of Taiwan with coordinates.**
(DOCX)

**S2 Table. Method detection limits (MDLs) and recovery (%) of PAHs analyzed in sediment samples in this study.**
(DOCX)

**S3 Table. Method detection limits (MDLs) and recovery (%) of PCBs, OCPs and PBDEs analyzed in sediment samples in this study.**
(DOCX)

**S4 Table. The quantitative ion and confirm ion in GC-MS/MS analysis for PCBs, OCPs and PBDEs in this study.**
(DOCX)

**S1 Protocol. Sampling station description and instrument analytical procedure.**
(DOCX)

## Acknowledgments

We acknowledge a grant of the Ministry of Science and Technology (MOST) of Taiwan for financial support through grants: MOST 104-2611-M-019- 004, MOST 105-2621-M-019 001, MOST 105-2918-I-019-001, MOST 106-2621-M-019- 001, and MOST 107-2621-M-019 001. We specially thank Mr. Chih-Ming Lin and Dr. Shao Hung Peng for their assistance during the field sampling. This paper is a contribution to the International Joint Laboratory between Lille University, France and National Taiwan Ocean University.

## Author Contributions

**Conceptualization:** Jing-O Cheng, Jiang-Shiou Hwang, Fung-Chi Ko.

**Data curation:** Shagnika Das, Andres Aria, Fung-Chi Ko.

**Formal analysis:** Shagnika Das.

**Funding acquisition:** Jiang-Shiou Hwang.

**Investigation:** Shagnika Das, Andres Aria.

**Methodology:** Jing-O Cheng, Fung-Chi Ko.

**Project administration:** Jiang-Shiou Hwang.

**Supervision:** Sami Souissi, Jiang-Shiou Hwang, Fung-Chi Ko.

**Validation:** Jing-O Cheng, Fung-Chi Ko.

**Visualization:** Fung-Chi Ko.

**Writing – original draft:** Shagnika Das, Jing-O Cheng.

**Writing – review & editing:** Fung-Chi Ko.

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
