## [Decision Letter · Decision Letter 0]

18 Oct 2019

PONE-D-19-27465

Occurrence and distribution of anthropogenic persistent organic pollutants in coastal
sediments and mud shrimps from the wetland of central Taiwan

PLOS ONE

Dear Dr. Ko,

Thank you for submitting your manuscript to PLOS ONE. After careful consideration, we
feel that it has merit but does not fully meet PLOS ONE’s publication criteria as it
currently stands. Therefore, we invite you to submit a revised version of the
manuscript that addresses the points raised during the review process.

We would appreciate receiving your revised manuscript by Dec 02 2019 11:59PM. When
you are ready to submit your revision, log on to https://www.editorialmanager.com/pone/ and select the 'Submissions
Needing Revision' folder to locate your manuscript file.

If you would like to make changes to your financial disclosure, please include your
updated statement in your cover letter.

To enhance the reproducibility of your results, we recommend that if applicable you
deposit your laboratory protocols in protocols.io, where a protocol can be assigned
its own identifier (DOI) such that it can be cited independently in the future. For
instructions see: http://journals.plos.org/plosone/s/submission-guidelines#loc-laboratory-protocols

We look forward to receiving your revised manuscript.

Kind regards,

Hans-Uwe Dahms, Ph.D.

Academic Editor

PLOS ONE

Journal Requirements:

2. We noticed you have some minor occurrence of overlapping text with the following
previous publication(s), which needs to be addressed: https://link.springer.com/article/10.1007%2Fs10661-008-0696-5,
https://link.springer.com/article/10.1007%2Fs11356-018-04113-x. In
your revision ensure you cite all your sources (including your own works), and quote
or rephrase any duplicated text outside the methods section. Further consideration
is dependent on these concerns being addressed.

3. Please amend your Methods section to clarify how the mud shrimp were collected,
stored, transported and sacrificed.

4. Please amend your Financial disclosure statement to declare sources of funding, or
state that the authors received no specific funding.

5. In your Methods section, please provide additional location information, including
geographic coordinates for the data set if available.

6. In your Methods section, please provide additional information regarding the
permits you obtained for the work. Please ensure you have included the full name of
the authority that approved the field site access and, if no permits were required,
a brief statement explaining why.

7.  We note that Figure 1in your submission contain [map/satellite] images which may
be copyrighted. All PLOS content is published under the Creative Commons Attribution
License (CC BY 4.0), which means that the manuscript, images, and Supporting
Information files will be freely available online, and any third party is permitted
to access, download, copy, distribute, and use these materials in any way, even
commercially, with proper attribution. For these reasons, we cannot publish
previously copyrighted maps or satellite images created using proprietary data, such
as Google software (Google Maps, Street View, and Earth). For more information, see
our copyright guidelines: http://journals.plos.org/plosone/s/licenses-and-copyright.

Additional Editor Comments (if provided):

The MS entitled "Occurrence and distribution of anthropogenic persistent organic
pollutants in coastal sediments and mud shrimps from the wetland of central Taiwan"
submitted to the journal PLoSone investigates the source and distribution of various
organic pollutants in mud shrimps and sediments around the industrial area, west
coast of central Taiwan. They concluded that concentrations of pollutants were
greater in samples collected near the industrial area than those from the
non-industrial locations and also revealed recent DDT inputs to the area.
Considering the aim, precise methodology and outcomes, the MS may be accepted for
publication with minor revisions.

The authors should consider the comments given by the reviewers below and return a
REVISED VERSION with an accompanying AUTHOR RESPONSE files until the 1st December
2019.

HANS-U. DAHMS, 17th October 2019

Reviewers' comments:

Reviewer's Responses to Questions

**Comments to the Author**

1. Is the manuscript technically sound, and do the data support the conclusions?

Reviewer #1: Yes

Reviewer #2: Yes

Reviewer #3: Yes

2. Has the statistical analysis been performed
appropriately and rigorously? 

Reviewer #1: Yes

Reviewer #2: Yes

Reviewer #3: Yes

3. Have the authors made all data underlying the
findings in their manuscript fully available?

Reviewer #1: Yes

Reviewer #2: Yes

Reviewer #3: Yes

4. Is the manuscript presented in an intelligible
fashion and written in standard English?

Reviewer #1: Yes

Reviewer #2: Yes

Reviewer #3: Yes

5. Review Comments to the Author

Reviewer #1: The MS entitled "Occurrence and distribution of anthropogenic persistent
organic pollutants in coastal sediments and mud shrimps from the wetland of central
Taiwan" submitted to the journal PLoSone investigates the source and distribution of
various organic pollutants in mud shrimps and sediments around the industrial area,
west coast of central Taiwan. They concluded that concentrations of pollutants were
greater in samples collected near the industrial area than those from the
non-industrial locations and also revealed recent DDT inputs to the area.
Considering the aim, precise methodology and outcomes, the MS may be accepted for
publication with minor revisions.

Specific comments:

Use degree symbols instead of using superscript of zero (e.g. line no. 97-98)

In this study mud shrimps were also collected to check the pollutants. There might be
water and/or sediment particles adhered on the body of the shrimps which may have
trace amount of pollutants. Before homogenizing them, how authors removed the
adhered water and/or sediment particles? Pls mention clearly in the methods section
(line 103)

Study area map (Figure 1) looks convincing; however authors should add the position
of the map and lat long details in the figure

Line 168-170: The sentence “Figure 2a shows the concentration of different
POPs....... season (January)” may be transferred to end of the paragraph, after line
no. 178

Line 172 & Tables: Pls define what is dw (dry weight?) in the MS

Line 218: change “and analyze relationships” to “and to analyze relationships”

Line 219: change “PC1 accounted for the major proportion” to “PC1 accounted major
proportion”

Line 251: change “may originate from” to “may be originated from”

Line 254: add a reference to the sentence “These results are consistent with our
previous findings”

Line 361: the formula mentioned here may be moved to the methods section

Reviewer #2: This manuscript presented a novel and distinctive study on the POPs in
coastal sediments and mud shrimps from the wetland of central Taiwan with supporting
results. The paper is suitable for publication with following corrections:

1. Add some of the most important quantitative results to the Abstract.

2. The discussion section should cite all the recent references.

3. Authors should carefully review the final resolution of the figures prior to
publication for better understanding of results. Especially Fig.4 is not clear.

4. Cite the source of future investigations in this study.

5. The authors are suggested to make changes according to the comments appended in
the manuscript’s PDF format.

6. Rectify grammatical errors to reach up to international standards.

7. Ensure that the references and whole manuscript is as per journal format. Remove
hyperlink from references.

Reviewer #3: This manuscript reported the concentrations of several POPs in sediment
and mud shrimps in a wetland of west coast of central Taiwan, China. It could
provide useful information for POPs distribution and risk assessment in wetlands.
Generally, it is well organized and written. However, the manuscript should be
revised before acceptance for publication. Details are as follows.

(1) The language should be improved. "At" should be used rather than "in" before
"station" or "site".

(2) "POPs" should be used over the whole text instead of "POP".

(3) Why chose the five stations? The authors should state their sampling design more
clearly. Besides, the ambient environment of the study area should be described.

(4) Page 15-16, the method for BAFs calculation should be moved to the Material and
methods part.

(5) It's OK to combine results and discussion together. However, the discussion is
very surficial. The authors should make deeper discussion, especially for the
mechanism of the distribution pattern of POPs.

(6) For Table 1, sediment type should be mentioned in each study.

(7) For Fig. 1, the longitude and latitude should be added on the map. The resolution
should be improved.

(8) Fig. 4 is difficult to read. Please improve the quality.

6. PLOS authors have the option to publish the peer
review history of their article (what does this mean?). If published, this will
include your full peer review and any attached files.

If you choose “no”, your identity will remain anonymous but your review may still be
made public.

**Do you want your identity to be public for this peer review?** For
information about this choice, including consent withdrawal, please see our
Privacy Policy.

Reviewer #1: No

Reviewer #2: No

Reviewer #3: Yes: Xiaoshou Liu

reviewer comments 14.10.19.docx
---

## [Author Response · Author response to Decision Letter 0]

12 Dec 2019

Letter from Editor:

PONE-D-19-27465

Occurrence and distribution of anthropogenic persistent organic pollutants in coastal
sediments and mud shrimps from the wetland of central Taiwan

PLOS ONE

Dear Dr. Ko,

Thank you for submitting your manuscript to PLOS ONE. After careful consideration, we
feel that it has merit but does not fully meet PLOS ONE’s publication criteria as it
currently stands. Therefore, we invite you to submit a revised version of the
manuscript that addresses the points raised during the review process.

We would appreciate receiving your revised manuscript by Dec 02 2019 11:59PM. When
you are ready to submit your revision, log on to https://www.editorialmanager.com/pone/ and select the 'Submissions
Needing Revision' folder to locate your manuscript file.

If you would like to make changes to your financial disclosure, please include your
updated statement in your cover letter.

To enhance the reproducibility of your results, we recommend that if applicable you
deposit your laboratory protocols in protocols.io, where a protocol can be assigned
its own identifier (DOI) such that it can be cited independently in the future. For
instructions see: http://journals.plos.org/plosone/s/submission-guidelines#loc-laboratory-protocols

• A rebuttal letter that responds to each point raised by the academic editor and
reviewer(s). This letter should be uploaded as separate file and labeled 'Response
to Reviewers'.

• A marked-up copy of your manuscript that highlights changes made to the original
version. This file should be uploaded as separate file and labeled 'Revised
Manuscript with Track Changes'.

• An unmarked version of your revised paper without tracked changes. This file should
be uploaded as separate file and labeled 'Manuscript'.

We look forward to receiving your revised manuscript.

Kind regards,

Hans-Uwe Dahms, Ph.D.

Academic Editor

PLOS ONE

Response: Thanks for the reminder. The following is the detailed response to the
journal requirements and the reviewers’ concerns and suggestions.

Journal Requirements:

Response: Yes, the revision has been adjusted to meet the standards of PLOS ONE’s
requirements.

2. We noticed you have some minor occurrence of overlapping text with the following
previous publication(s), which needs to be addressed: https://link.springer.com/article/10.1007%2Fs10661-008-0696-5,
https://link.springer.com/article/10.1007%2Fs11356-018-04113-x. In
your revision ensure you cite all your sources (including your own works), and quote
or rephrase any duplicated text outside the methods section. Further consideration
is dependent on these concerns being addressed.

Response: Yes, thanks for the comments. The references have been cited, and the
duplicated text have been rephrased in the revision.

3. Please amend your Methods section to clarify how the mud shrimp were collected,
stored, transported and sacrificed.

Response: Collection and treatment of mud shrimp samples in methods section are
updated with the cited references.

4. Please amend your Financial disclosure statement to declare sources of funding, or
state that the authors received no specific funding.

Response: The financial disclosure statement has been added to declare sources of
funding.

5. In your Methods section, please provide additional location information, including
geographic coordinates for the data set if available.

Response: Yes, the information of sampling cites can be found in the supplementary
data in Table S1.

6. In your Methods section, please provide additional information regarding the
permits you obtained for the work. Please ensure you have included the full name of
the authority that approved the field site access and, if no permits were required,
a brief statement explaining why.

Response: The permits for sampling mud shrimps is not required. The information is
added in the revision text (Line 101-102).

7. We note that Figure 1in your submission contain [map/satellite] images which may
be copyrighted. All PLOS content is published under the Creative Commons Attribution
License (CC BY 4.0), which means that the manuscript, images, and Supporting
Information files will be freely available online, and any third party is permitted
to access, download, copy, distribute, and use these materials in any way, even
commercially, with proper attribution. For these reasons, we cannot publish
previously copyrighted maps or satellite images created using proprietary data, such
as Google software (Google Maps, Street View, and Earth). For more information, see
our copyright guidelines: http://journals.plos.org/plosone/s/licenses-and-copyright.

Response: The image in figure 1 was generated from DIVA-GIS, using free spatial data
and QGIS version 2.6, and is not copyrighted. 

Response: Thanks for your comments. The captions for the supporting information files
have been added at the end of the manuscript (line 662-663).

Additional Editor Comments (if provided):

The MS entitled "Occurrence and distribution of anthropogenic persistent organic
pollutants in coastal sediments and mud shrimps from the wetland of central Taiwan"
submitted to the journal PLoSone investigates the source and distribution of various
organic pollutants in mud shrimps and sediments around the industrial area, west
coast of central Taiwan. They concluded that concentrations of pollutants were
greater in samples collected near the industrial area than those from the
non-industrial locations and also revealed recent DDT inputs to the area.
Considering the aim, precise methodology and outcomes, the MS may be accepted for
publication with minor revisions.

The authors should consider the comments given by the reviewers below and return a
REVISED VERSION with an accompanying AUTHOR RESPONSE files until the 1st December
2019.

HANS-U. DAHMS, 17th October 2019

Response: Thanks so much for the comments. The manuscript has been revised and
resubmitted to PLos One for publication.

Reviewers' comments:

Reviewer's Responses to Questions

Comments to the Author

1. Is the manuscript technically sound, and do the data support the conclusions?

Reviewer #1: Yes

Reviewer #2: Yes

Reviewer #3: Yes

Response: Thanks.

2. Has the statistical analysis been performed appropriately and rigorously? 

Reviewer #1: Yes

Reviewer #2: Yes

Reviewer #3: Yes

Response: Thanks.

3. Have the authors made all data underlying the findings in their manuscript fully
available?

Reviewer #1: Yes

Reviewer #2: Yes

Reviewer #3: Yes 

Response: Thanks. 

4. Is the manuscript presented in an intelligible fashion and written in standard
English?

Reviewer #1: Yes

Reviewer #2: Yes

Reviewer #3: Yes

Response: Thanks.

5. Review Comments to the Author

Reviewer #1: The MS entitled "Occurrence and distribution of anthropogenic persistent
organic pollutants in coastal sediments and mud shrimps from the wetland of central
Taiwan" submitted to the journal PLoSone investigates the source and distribution of
various organic pollutants in mud shrimps and sediments around the industrial area,
west coast of central Taiwan. They concluded that concentrations of pollutants were
greater in samples collected near the industrial area than those from the
non-industrial locations and also revealed recent DDT inputs to the area.
Considering the aim, precise methodology and outcomes, the MS may be accepted for
publication with minor revisions.

Response: Thanks for your suggestions. The minor revisions of this manuscript had
been completed.

Specific comments:

Use degree symbols instead of using superscript of zero (e.g. line no. 97-98)

In this study mud shrimps were also collected to check the pollutants. There might be
water and/or sediment particles adhered on the body of the shrimps which may have
trace amount of pollutants. Before homogenizing them, how authors removed the
adhered water and/or sediment particles? Pls mention clearly in the methods section
(line 103)

Response: The degree symbols have been corrected (lines 92, 93, and 100). The
description has been rewritten in method section (lines 97-102).

Study area map (Figure 1) looks convincing; however authors should add the position
of the map and lat long details in the figure

Response: The location and characteristics of sampling cites including latitudes and
longitudes are described in the supplementary data at Table S1.

Line 168-170: The sentence “Figure 2a shows the concentration of different
POPs....... season (January)” may be transferred to end of the paragraph, after line
no. 178

Response: Thanks for the suggestion. This sentence has been moved to the end of this
paragraph (lines 180-182).

Line 172 & Tables: Pls define what is dw (dry weight?) in the MS

Response: Yes, the dw has been defined as dry weight at the first time appear in the
text (line 173).

Line 218: change “and analyze relationships” to “and to analyze relationships”

Response: Thanks for the suggestion. It has been approved (line 219).

Line 219: change “PC1 accounted for the major proportion” to “PC1 accounted major
proportion”

Response: Thanks for the suggestion. It has been approved (line 220).

Line 251: change “may originate from” to “may be originated from”

Response: Thanks for the suggestion. It has been approved (line 247).

Line 254: add a reference to the sentence “These results are consistent with our
previous findings”

Response: The description has been rewritten.

Line 361: the formula mentioned here may be moved to the methods section

Response: Thanks for the suggestions. The formula has been moved to the methods
section.

Reviewer #2: This manuscript presented a novel and distinctive study on the POPs in
coastal sediments and mud shrimps from the wetland of central Taiwan with supporting
results. The paper is suitable for publication with following corrections:

1. Add some of the most important quantitative results to the Abstract.

Response: Thanks for the suggestion. We previously included the bioaccumulation
factors for DDTs and PCBs. We have added in quantitative results for PAH
concentrations in heavily contaminated site compared to less contaminated sites.

2. The discussion section should cite all the recent references.

Response: Thanks for the suggestions. The discussion section has been rewritten.

3. Authors should carefully review the final resolution of the figures prior to
publication for better understanding of results. Especially Fig.4 is not clear.

Response: Thanks for the comments. Figure 4 has been corrected to 2D, instead of 3D
providing a clear resolutions.

4. Cite the source of future investigations in this study.

Response: Thanks for the suggestion. Citations have been reviewed and updated.

5. The authors are suggested to make changes according to the comments appended in
the manuscript’s PDF format.

Response: Thanks for the suggestions. The revision has been improved according your
valuable suggestions.

6. Rectify grammatical errors to reach up to international standards.

Response: The manuscript has been carefully polished by correcting the grammatical
errors.

7. Ensure that the references and whole manuscript is as per journal format. Remove
hyperlink from references.

Response: The reference format has been adjusted to match the publishing standard of
PLos One.

Reviewer #3: This manuscript reported the concentrations of several POPs in sediment
and mud shrimps in a wetland of west coast of central Taiwan, China. It could
provide useful information for POPs distribution and risk assessment in wetlands.
Generally, it is well organized and written. 

However, the manuscript should be revised before acceptance for publication. Details
are as follows.

(1) The language should be improved. "At" should be used rather than "in" before
"station" or "site".

Response: Thanks for the suggestions. The preposition “in” has been replaced by “at”,
before sampling stations or sites as suggestion.

(2) "POPs" should be used over the whole text instead of "POP".

Response: Thanks for the comments. It has been adjusted to consistently use
“POPs”.

(3) Why chose the five stations? The authors should state their sampling design more
clearly. Besides, the ambient environment of the study area should be described.

Response: The characteristics of the sampling sites are described in the Table S1 in
the supplementary data.

(4) Page 15-16, the method for BAFs calculation should be moved to the Material and
methods part.

Response: Thanks for the suggestions. The BAF equation has been moved to the
Materials and methods section.

(5) It's OK to combine results and discussion together. However, the discussion is
very surficial. The authors should make deeper discussion, especially for the
mechanism of the distribution pattern of POPs.

Response: Results and discussion section has been rewritten and updated.

(6) For Table 1, sediment type should be mentioned in each study.

Response: The sediment information in Table 1 are from the coastal or estuarine
areas.

(7) For Fig. 1, the longitude and latitude should be added on the map. The resolution
should be improved.

Response: Yes, the Figure 1 have been updated with longitude and latitude.

(8) Fig. 4 is difficult to read. Please improve the quality.

Response: Yes, the quality of Figure 4 has been improved.

6. PLOS authors have the option to publish the peer review history of their article
(what does this mean?). If published, this will include your full peer review and
any attached files.

If you choose “no”, your identity will remain anonymous but your review may still be
made public.

Do you want your identity to be public for this peer review? For information about
this choice, including consent withdrawal, please see our Privacy Policy.

Reviewer #1: No

Reviewer #2: No

Reviewer #3: Yes: Xiaoshou Liu

to reviewers.docx
---

## [Editor Report · Decision Letter 1]

18 Dec 2019

Occurrence and distribution of anthropogenic persistent organic pollutants in coastal
sediments and mud shrimps from the wetland of central Taiwan

PONE-D-19-27465R1

Dear Dr. Ko,

We are pleased to inform you that your manuscript has been judged scientifically
suitable for publication and will be formally accepted for publication once it
complies with all outstanding technical requirements.

With kind regards,

Hans-Uwe Dahms, Ph.D.

Academic Editor

PLOS ONE

Additional Editor Comments (optional):

Dear Prof. KO,

I saw that all the issues raised by the REVIEWERS were implemented

in your revised version. I am now glad to tell you the ACCEPTANCE of

your contribution to PONE.
---

## [Editor Report · Acceptance letter]

26 Dec 2019

PONE-D-19-27465R1 

Occurrence and distribution of anthropogenic persistent organic pollutants in coastal
sediments and mud shrimps from the wetland of central Taiwan 

Dear Dr. Ko:

I am pleased to inform you that your manuscript has been deemed suitable for
publication in PLOS ONE. Congratulations! Your manuscript is now with our production
department. 

With kind regards,

on behalf of

Dr. Hans-Uwe Dahms 

Academic Editor

PLOS ONE